# Flood Mitigation in Urban Areas through Deep Aquifer Recharge: The Case of the Metropolitan Area of Guadalajara

**DOI:** 10.3390/ijerph19063160

**Published:** 2022-03-08

**Authors:** Luis Ignacio Vanegas-Espinosa, David Vargas-del-Río, Gabriela Ochoa-Covarrubias, Alejandro Luis Grindlay

**Affiliations:** 1Department of Habitat and Urban Development, Instituto Tecnológico y de Estudios Superiores de Occidente, ITESO, The Jesuit University of Guadalajara, Tlaquepaque 45604, Mexico; luisvanegas@iteso.mx (L.I.V.-E.); davidvar@iteso.mx (D.V.-d.-R.); ochoagabriela@iteso.mx (G.O.-C.); 2Department of Urban and Regional Planning, University of Granada, 18071 Granada, Spain

**Keywords:** surface waterproofing, deep injections wells, artificial aquifer recharge, flood reduction, sustainable urban drainage systems, rainwater utilization

## Abstract

The Metropolitan Area of Guadalajara (MAG) experiences water shortage and overexploitation of aquifers. In addition, it suffers from seasonal flooding that is channeled towards inadequate sanitary drainage, creating a strong negative environmental impact. These problems are rooted in the waterproofing of the urban surface. Many cities around the world have used deep injection wells to recover aquifers and remove surface waters. Certain geohydrological conditions are required for the implementation of these deep injection wells, deeper than 30 m, such as significant surface runoff, acceptable water quality for infiltration, considerable depth in the phreatic levels, and good subsoil permeability. All of these conditions exist in the MAG or could be achieved without significant investment. An assessment is presented exploring the viability for a solution based on this technology, as a strategy to recover aquifers and reduce flooding. The first step was to identify, through map algebra, a micro-basin suitable for this technology. Then, mean runoff volumes were obtained and a stratigraphic profile was carried out based on 19 standard penetration tests (SPT). With these data, a numerical simulation of deep injection wells of different dimensions was performed, providing recommendations for a solution based on these calculations. The results show that both problems can be solved with this relatively simple and cheap technology supporting public health.

## 1. Introduction

Water availability is a growing issue worldwide. An assessment of analyses showed that between 1995 and 2025 the zones of severe hydric stress will have increased by 2.2 million km^2^, encompassing a total area of 38.6 million km^2^, affecting 4000 million people. According to this assessment, the main reason for this is the inadequate management of the water resources. This situation is exacerbated in developing countries, where minimal volumes of water are treated and recycled. The projected scenario is one of a rapid reduction in both quantity and quality of the available water, increasing water emergencies, more conflicts related to rights and access to water resources, and a decrease in industrial, economic, and agricultural development, as well as the inherent negative environmental impact [1,2,3]. The leading solution calls for a more efficient use of water resources. However, it was predicted that this solution will not allow for the increasing water demand due to population and economic growth [3].

In Mexico, just as predicted, a water scarcity scenario is taking place, where more than 15% of the aquifers are overexploited and represent the main source of provision for 60% of the population, with conditions of increasing semiaridity in their central and northern parts [4,5]. The Metropolitan Area of Guadalajara (MAG) is currently in this situation and is facing a state of growing hydrological distress [6,7]. However, it is the intent of this paper to demonstrate that engineering and better management could provide a solution to the root of the problem. The paper explores an alternative view and proposes methods that can be applied in many other urban zones.

In the context described in the next section, a pattern of urban growth caused by immediate necessity has created the hydrological risk. As the city has grown, buildings and roadways have covered and waterproofed the surface, disrupting the hydrological cycle and generating severe environmental problems. This disconnection produces two effects. Above ground, the original rainwater drainage system is exceeded and the city floods periodically; in just 2015 this caused nine deaths and economic losses of approximately USD 62.5 million [8,9]. Below ground, the aquifers do not receive sufficient recharge. As the advance of urbanization also intensifies the water requirements, the groundwater system becomes overexploited, the public services deteriorate, and the hydrology, as a whole, declines [10]. Finally, the situation requires more investment in infrastructure and hydric megaprojects, eventually creating new problems in the city’s periphery.

Sustainable urban drainage systems (SUDS) propose to counteract this negative feedback loop by connecting the surface with the aquifers. They are designed to infiltrate the runoff in order to reduce floods and utilize the water in drought periods. The correct implementation of this approach requires water pretreatment technologies to avoid pollutants, hydraulic infrastructure to channel runoff without destabilizing the surrounding ground, and artificial recharge procedures that promote infiltration [11]. The elements that control the design of the entire system are the surface runoff, the stratigraphy, and the area available for infiltration. This produces three different methods of recharge: surface recharge systems, deep recharge systems, and mixed recharge systems. Surface recharge systems are designed to increase the soil–water contact surface (through infiltration trenches, artificial ponds, or riverbed conditioning), and these constitute the best option for free-aquifer recharge when there is enough surface area available. Deep recharge systems include vertical wells, infiltration galleries, and radial drains. These are more expensive to build and maintain and require better infiltration water quality; however, they need a smaller area to perform infiltration, which is a relevant characteristic in urban contexts. In addition, they allow the recharging of confined aquifers. Finally, there are mixed recharge systems that include elements from both surface and deep systems [12]. A recharge system for the MAG is proposed based on vertical deep wells, deeper than 30 m, since less urban area is required, and due to their efficiency in recharging confined and semi-confined aquifers.

Most of the water regulatory organizations consider this solution as an emergent, challenging, risky, and unviable technology [13]. However, this view is not shared by a growing group of experts in hydrogeology and geotechnics, who recognize its potential contribution to increasing water provision and ensuring hydric safety and water quality [12,13]. As an example, the cities of Dusseldorf and Budapest depend entirely on artificially recharged underground waters. The technology has certainly been widely tested [2,14,15].

## 2. Study Area

The MAG is the third largest metropolitan area by population in Mexico and it covers an urbanized area of 679 km^2^. It is inhabited by 5.2 million people who experience seasonal water shortages resulting in aquifer overexploitation, even though precipitation levels and nearby water sources are plentiful. The city is supplied by Chapala Lake, the largest in Mexico. It also obtains water from deep wells and the Calderon Dam [16]. Altogether, these sources provide a volumetric flux of 9.5 m^3^/s. However, this amount is significantly less than the 13.06 m^3^/s established by the institutions in charge of managing the water resources [16,17]. As a consequence of this shortfall, new dams and hydraulic infrastructure projects are being developed.

Nevertheless, there is one fundamental omission in this analysis: the city is not utilizing the mean annual rainfall of 942 mm and, according to the World Meteorological Organization, this precipitation should be enough to supply the city’s population [18]. The accuracy of this argument may be demonstrated simply by multiplying this precipitation by its total area and then dividing it by the number of seconds in one year. The result is a potential volumetric flux of 81.66 m^3^/s. In other words, the MAG needs to catch only 16% of the annual precipitation it receives in order to become self-sufficient in its water requirements. It would be more beneficial to adopt this as a solution rather than building more infrastructure (with its implicit negative environmental impact).

Hydrologically, the MAG is bordered by the volcanic complex La Primavera (VCLP). The geological stratigraphy comprises a bedrock, covered by alternated strata of tuff, lapilli, and volcanic ash. These strata extend beneath most of the city and their thickness decreases outward from the VCLP. In a radius of three to five kilometers, the thickness exceeds one hundred meters, and is between twenty to fifty meters thick at the city center [19]. The permeability of these strata is 0.0022–0.0043 m/s for lapilli, 0.0015–0.0029 for volcanic ash, around 0.0008 for tuff, and lower than 10^−7^ for the bedrock [20]. These differences in permeability result in a sequence of semiconfined aquifers of high porosity (20% to 40%), along with a macrosystem that behaves like a free aquifer [19].

A large normal fault divides this macrosystem and demarcates two overexploited aquifers that provide 30% of the water [16]. One is the Atemajac aquifer, over which most of the city is located, and which is undergoing a static level decrease of 3.5 m/year and drawdown levels that have reached 150 m. The other is the Toluquilla aquifer, with a static level decrease of 0.5 to 1.5 m/year and drawdown levels that are between 75 to 100 m, worsening towards the urbanized area [18]. Both aquifers experience significant deficits: 11,000,000 m^3^ in the first case and 72,000,000 m^3^ in the second [4,16] (see Figure 1).

## 3. Materials and Methods

An assessment was developed to evaluate the viability of a deep well artificial aquifer recharge system in the MAG. The design of this system was carried out through the study of its potential flood and soil sealing reduction via subsoil infiltration in one urban micro-basin; that is, a small topographic region in which all water drains to a common area and is part of a larger-scale basin. The assessment was achieved by considering the infiltration changes under different deep-well dimensions, and average subsoil and hydrological conditions of the micro-basin.

Three types of analysis were performed for this viability assessment:Hydrological study of the MAG to calculate the amount of runoff that needs to be evacuated.Mean geohydrological conditions of the ground below the chosen micro-basin, to evaluate the capacity of subsoil for receiving such runoff.Numerical estimation of the infiltration of theoretical wells of different sizes to provide for an adequate design of the solution.

The hydrological study examined the MAG in order to select a micro-basin with seasonal flooding suitable for this viability assessment. First, a survey of the MAG was undertaken with map algebra in a Geographic Information System, using the information on aquifers, public usage wells, current and pre-existing streams (i.e., those that have been eliminated due to urbanization), terrain slopes, topography on a 1:10,000 scale, and flood records (Table 1). Since the subsoil condition is expected to be homogeneous in this geological context, the main criterion for selecting the micro-basin was the possibility of alleviating the persistent floods (see Figure 2).

Once the micro-basin study area was chosen, its surface runoffs and maximum volumetric water flux were evaluated using the rational method (Table 1), using the following Equation (1):(1)$$Q_{p}=\mathrm{Maximum}\;\mathrm{flux}\;\left(m^{3}/s\right)=0.278C_{p}\mathrm{iA}$$
where
Cp = Weighted runoff coefficient according to the drained surface.i = Mean rainfall intensity on the micro-basin during the concentration time (tcs) for a return period of 5 years.A = Basin area (km^2^).


The Intermunicipal System of Potable Water and Sewerage Services (SIAPA after its acronym in Spanish) requires the use of this process to calculate maximum surface runoffs and to design adequate sanitary infrastructure. This approach considers equal precipitation on the entire micro-basin for a specific time [17]. The consideration at this scale could result in an overestimation of the surface runoffs. However, this is the prevailing norm, also constraining this viability assessment, and it was assumed as a security factor.

After the maximum volumetric water flux was obtained, the geohydrological recognition was examined from 19 standard penetration tests (SPT) provided by geotechnical engineering companies. After this revision, the stratigraphy provided by the geotechnical engineering company Sandstorm Gam was selected (see Figure 3). The bedrock and phreatic level in the area were obtained in other studies. The bedrock was located according to the work of Zamudio and Gómez [22] as well as the reported stratigraphy of the existing extraction wells [19]. Similarly, the phreatic level depth was established according to the reported levels of extraction, in the wells operated by SIAPA and the levels reported in GEOEX-SIAPA [19]. Once the mean stratigraphic profile for the micro-basin was defined, each stratum was associated with a value of saturated hydraulic conductivity (Ksat).

## 4. Results

### 4.1. Hydrological Study of the MAG

The hydrological study of the MAG resulted in the El Chicalote micro-basin being chosen, as it is a natural riverbed modified by urban layout. Analysis of this area allows the conflicts caused by the breakdown of the hydrological cycle due to the waterproofing of the surface to be shown (see Figure 1). This choice is based on two main factors. First, the lower part of this basin is an infamous flood area during the rainy season (the three points with the highest number of flood reports are located within the micro-basin: the Parque de la Liberación area, Calzada Independencia at its intersections with Revolución and Juárez avenues, and, finally, the intersection of Mariano Otero and López Mateos avenues where Plaza del Sol is located). Second, this micro-basin extends over the upper part of the hydrogeological system of the city, on its western boundary. Surface runoff in the east is more contaminated as it flows over the paved surface. Therefore, water for infiltration is cleaner on the western boundary and this may be a good place to start a future aquifer recharge plan throughout the city (see Figure 2).

Figure 2 shows the chosen micro-basin along with information on the recurrent flood areas, drainage extraction wells, and infiltration trenches/channels. This watershed fed a pre-existing stream called Arroyo del Chicalote. As can be observed, this area undergoes recurrent floods (orange dots) caused by annual runoff (solid blue line), clearly associated with the urbanization over a pre-existing waterbed (blue dotted line).

Given its particular dimensions it is possible to use the rational method. This empirical Formulation (2) allows the maximum volumetric flux (m^3^/s) to be calculated in a micro-basin such as this. The general parameters for calculation are presented and explained as follows:(2)$$Q_{p}=\mathrm{Maximum}\;\mathrm{flux}\;\left(m^{3}/s\right)=0.278C_{p}\mathrm{iA}\;=0.278\times 0.65\times 56.238\;\mathrm{mm}/h\times 11.569\;{\mathrm{km}}^{2}=117.57\;m^{3}/s$$
where
$$A=\mathrm{Basin}\;\mathrm{area}\;({\mathrm{km}}^{2})=11.569\;{\mathrm{km}}^{2}$$
$$C_{p}=\mathrm{Weighted}\;\mathrm{runoff}\;\mathrm{coefficient}\;\mathrm{according}\;\mathrm{to}\;\mathrm{the}\;\mathrm{drained}\;\mathrm{surface}:$$
$$C_{p}=\frac{\Sigma \left(\mathrm{Ci}\times \mathrm{Ai}\right)}{\Sigma \mathrm{Ai}}=0.65$$
where
Ci= Partial runoff coefficient according to the surface and obtained from the rational method coefficient tables (the difference between streets’ and residential houses’ runoff coefficients were disregarded, as residential houses’ maximum runoff value matches the street’s minimum value).
$$C_{\mathrm{parks}\;\mathrm{and}\;\mathrm{green}\;\mathrm{areas}}=0.25.$$
$$C_{\mathrm{residential}}=0.70.$$
Ai=Partial area correspondent to each surface.
$$A_{\mathrm{parks}\;\mathrm{and}\;\mathrm{green}\;\mathrm{areas}}=1.28\;{\mathrm{km}}^{2}$$
$$A_{\mathrm{residential}}=10.29\;{\mathrm{km}}^{2}$$
i = Mean rainfall intensity on the micro-basin during the concentration time (tcs) for a return period of 5 years, obtaining an interpolated value of 56.238 mm/h according to the rational method coefficient tables.
(3)$$t_{\mathrm{cs}}=0.0003245{\left(\frac{L}{\sqrt{S}}\right)}^{0.77}=0.0003245{\left(\frac{6790\;m}{\sqrt{0.0447\;m/m}}\right)}^{0.77}=0.958\;h=57.48\;\min$$
where
L=Length of the main watershed=6790 m.
$$S=\mathrm{Mean}\;\mathrm{slope}\;\mathrm{of}\;\mathrm{the}\;\mathrm{watershed}\;=\frac{\mathrm{Maximum}\;\mathrm{height}-\mathrm{Minimum}\;\mathrm{height}}{L}=\frac{1894\;m-1590\;m\;}{6790\;m}=0.0447\;m/m$$

### 4.2. Mean Geohydrological Conditions of the Ground below the Chosen Micro-Basin

As described above, the stratigraphy of the ground below the chosen micro-basin was determined from other studies. The SPT selected describe the stratigraphy to 54 m by using the Unified Soil Classification System (USCS); it is displayed in Figure 3. As can be seen, it comprises silty sands (SM), or ashes under its volcanic designation, alternated with poorly graded sands (SP), or lapilli.

Through this comparison, the bedrock was established for a depth of 60 m and deeper than 60 m for the phreatic level. The extraction wells’ information was obtained through the Institute of Transparency, Public Information and Personal Data Protection (ITEI) resolution no. 4284217 sent to SIAPA on 9 October 2017. The locations of the extraction wells chosen for the study area are shown in Figure 2.

### 4.3. Numerical Estimation of the Infiltration of Potential Wells

Darcy’s permeability or hydraulic conductivity were determined from the work of Zamudio, Vargas, and Ochoa [20], who established coefficients for the MAG soils according to the USCS through empirical formulations that allow permeability values to be obtained from particle size analysis. These permeability coefficients are consistent with those reported by GEOEX-SIAPA [18,19]. In the case of basalt, the values published in Sanders [24] for unaltered basalt were used. As a result,
$${\mathrm{Ksat}}_{\mathrm{SP}}=\mathrm{Poorly}\;\mathrm{graded}\;\mathrm{sands}\;\mathrm{permeability}\;\left(\mathrm{pumice}\;\mathrm{sands}\right)=2.7\times 10^{-3}\;m/s.$$
$${\mathrm{Ksat}}_{\mathrm{SM}}=\mathrm{Silty}\;\mathrm{sand}\;\mathrm{permeability}=8.0\times 10^{-4}\;m/s$$
$${\mathrm{Ksat}}_{\mathrm{BASALT}}=\mathrm{Basalt}\;\mathrm{permeability}=1.16\times 10^{-8}\;m/s$$

By applying finite element modeling (FEM) with the mean stratigraphic profile described above, a numerical estimation was made for the maximum infiltration capacity of wells of different sizes (Figure 4). FEM is a numerical approach that allows for solutions where partial differential equations that describe a physical process are associated with dependent variables, as is the case with the mean expected water flux through a known stratigraphy of porous, non-saturated soils, because of the infiltration of a specified deep well. The analysis tool SEEP/W of the GEOSTUDIO platform was used to achieve this numerical simulation. The first step in SEEP/W is to define the type of analysis, which can be either static or transient. In the former, analysis only takes place when there are no changes in the flow conditions over time and not with the variation of time, as is the case in transient analysis. Next, the display of the model is defined, which can be one-dimensional, two-dimensional, or axisymmetric. Axisymmetric means that the plane drawn on the model is symmetric around an axis. In this case, an axisymmetric transient analysis was chosen. Once the type of analysis is defined, it is necessary to generate the geometry of the model, in which the stratigraphic profile was included. Thus, the type of materials is defined since they can be saturated or saturated/unsaturated soils. Due to the nature of the project, the changes in the permeability and in the flow of water in the soils need to be established, and so saturated/unsaturated soils were chosen. Four characteristics of soil need to be considered to be able to solve a flow problem: the hydraulic conductivity function, the volumetric water content function, the anisotropy ratio, and the angle of rotation. Finally, the boundary conditions are defined. In this case, a total load vs. time function was defined, where a specific load (H) was established throughout the deep injection well. For this, four different functions were defined where H was kept constant in each of the functions, but it was modified between the functions to observe the differences in the flow rate and total accumulated volume. That is, if the value of H increases, it is infiltrating into higher strata each time and vice versa. For example, referring to Figure 4, if it is indicated that H = 35, it will only infiltrate from elevation 25 to elevation 35, that is, only in the first stratum. The time was taken from 14 days constant in all cases. This method entails the following steps (GEO-SLOPE International Ltd., Calgary, Alberta, Canada, 2018):Discretization of the domain in finite elements.Function selection to describe the way the main variable changes through each component.Definition of a constitutive equation.Equations derivation.Global equations assemblage and frontier conditions modification.Solution of the global equations.

Once the global equations were calculated, a general spatial and temporal description of the main variable through each domain was obtained, in this case, the water flowing through a porous non-saturated soil. The analysis was carried out by considering a transitory water flux in an axisymmetric soil geometry that may or may not be saturated, employing the method proposed by Van Genuchten [25]. The volumetric water content (θw) was determined using the following equation:$$\theta w=\theta \mathrm{res}+\frac{\theta \mathrm{sat}-\theta \mathrm{res}}{{\left[1+{\left(\alpha {}^{\prime }\varphi \right)}^{n}\right]}^{m}}$$
where
$\alpha {}^{\prime }$, n, m = Adjustment parameters of the curves controlling the function’s shape.φ = Matric suction (kPa).θsat = Volumetric water content in saturated state.θres = Residual volumetric water content.


In addition, permeability Kw(φ) in saturated or non-saturated state was determined based on the corresponding saturated permeability values (Ksat_(SP,) Ksat_SM, Ksat_BASALT) according to the stratigraphy:$$\mathrm{Kw}\left(\varphi \right)=\mathrm{Ksat}\frac{{\left\{1-{\left(\alpha {}^{\prime }\varphi \right)}^{n-1}{\left[1+{\left(\alpha {}^{\prime }\varphi \right)}^{n}\right]}^{-m}\right\}}^{2}}{{\left[1+{\left(\alpha {}^{\prime }\varphi \right)}^{n}\right]}^{\frac{m}{2}}}$$
where
$$K_{\mathrm{sat}}=\mathrm{Permeability}\;\mathrm{in}\;\mathrm{saturated}\;\mathrm{state}.$$

Finally, the wells’ piezometric head conditions were defined. Therefore, as the purpose was to determine the wells’ maximum infiltration capacity, flooding conditions were considered; that is, full head over the whole well for a relatively extended period, established as 14 days.

While simulating the maximum infiltration volumetric fluxes caused by deep wells of different geometries, changes in the diameter resulted in negligible modifications in the infiltration volumes. Therefore, this variable was set at 1 m, according to construction and maintenance criteria. In contrast, significant differences were observed in maximum infiltration volumetric fluxes when simulating changes in the wells’ depth, as can be observed in Table 2 and Figure 5. This is the case because the water can permeate towards poorly graded sand strata, with higher water conductivity, while the hydraulic head increases. As can be seen, during the first 12 h, the 35 m depth wells can infiltrate a mean value of 0.42 m^3^/s, and this value increases to 1.85 m^3^/s when the depth is extended to 65 m. These are high values for potential flood mitigation. Furthermore, for 14 days during the rainy season, wells can inject from 50,000 to 285,000 m^3^ for 35 to 65 m. As the area of the aquifer is 649.97 km^2^ with porosity of 30%, these values represent a recharge contribution of 0.23 to 1.43 mm/well. These are high values for potential water storage and aquifer recharge. 

## 5. Discussion

The results of this assessment indicate that the effects of soil sealing and flooding may be reduced using deep injection wells, also helping to alleviate water scarcity in the MAG, as a sustainable urban water management strategy proposed and developed in other Mexican areas and in the world [26,27,28,29,30,31]. With respect to the previously mentioned systems of Dusseldorf and Budapest, in the case of the long-term system of Dusseldorf (Germany), riverbank filtration has been used for over a century as a first natural treatment step, and its population is supplied with treated bank filtrate by four waterworks, including vertical wells of around 20 m and horizontal collector wells [32]. In the case of Budapest (Hungary), it is remarkable that the wells at Csepel Island site are situated in an aquifer of 15 m maximum thickness for its water production of 400,000 m^3^/d [33]. In these cases, the extraction wells are shallower than the proposed infiltration wells.

These proposed wells are particularly advantageous in this hydrogeological context because water penetrates through more permeable strata, with a higher piezometric head. It is worth mentioning that this proposal is different from the recharge strategy currently implemented in the MAG, which consists of vertical shallow wells, commonly known as absorption wells. Although they are cheap and have been proven capable of infiltrating significant water volumes, they do have certain drawbacks: (1) they have a tendency to quickly become clogged and in some cases are prone to phreatic pollution since they do not include a pretreatment system in their design; (2) underground erosion and damage to nearby buildings when built on landfills; (3) limited infiltration capacity due to their reduced dimension. The deep injection well solution proposed in this work is a different matter. The infiltration of landfills must be avoided, and water treatment must also be considered to comply with an environmental normativity—for example, the NOM-015-CONAGUA-2007 “Aquifer’s artificial water infiltration. Characteristics and specifications of water and waterworks”. An inexpensive option for achieving this is to prevent the infiltration of the first annual rainfalls.

In line with these results, a viable solution for the runoff discharges and infiltration requirements of the micro-basin El Chicalote is to build wells of 1 m width and deeper than 55 m, avoiding infiltration in landfills and other earthworks. These wells will absorb more than 1 m^3^/s, mitigating flooding during elevated precipitation episodes, and accumulating a water volume proportional to the attributes of their location, contributing to aquifer recovery. According to the calculated maximum volumetric flux in this micro-basin (117.57 m^3^/s), one of the proposed deep injection wells would be capable of infiltrating more than 0.9% of the maximum runoff that could take place every five years. In other words, it would be necessary to build approximately 110 wells along the micro-basin to infiltrate the calculated runoff volumes and prevent the floods. Regarding the location of the wells, it is recommended that they be distributed along the basin, in order to infiltrate the runoff as it flows down. This well system may reduce their infiltration efficiency after a continuous period of precipitation, or once the aquifer is recovered. However, considering the present overexploitation of the aquifer and the option of extracting the water for supplying the city during the drought season, their efficiency will probably last for several years.

A simple cost–benefit analysis can assess the economic viability of this solution. As previously mentioned, the flood problem caused financial losses of approximately USD 62.5 million in 2015 [9]. In contrast, the cost of constructing one of the recommended deep wells is around USD 125,000, according to an estimate from a drilling company. These rough values suggest the financial outlay caused by the inundations of one year may be used, instead, for constructing 500 wells and solving the floods in one of the most exposed areas in the MAG, whilst helping in the recovery of the overexploited aquifer. Therefore, this technology is viable even without considering the environmental and social externalities, such as future water scarcity and aquifer overexploitation. Because of the geological context of the MAG, the results obtained from this assessment can be transferred to other micro-basins. Deeper or shallower wells may be used, according to the subsoil conditions and the infiltration requirements (see Figure 6).

This recharge plan can support public health in at least three ways:(1)Providing a larger quantity of higher quality water for consumption and other purposes. Under the present circumstances, because of water leaks in the sewer system and overexploitation, contamination tends to concentrate.(2)Enhancing the quality of the environment in regularly flooded areas, because the rain carries the pollutants and concentrates them in the low-level areas, which are regularly flooded.(3)Contributing to the reduction of respiratory illness, because water is also a thermal regulator; recharging the aquifer implies fewer temperature fluctuations.

## 6. Conclusions

The main purpose of this assessment was to explore the viability of a deep well artificial aquifer recharge system in the MAG. According to the results of this study, the context of the MAG is suitable for the implementation of a deep-well artificial recharge system. However, certain considerations need to be noted. The rational method was required for the calculation of maximum surface runoffs and for the design of a sanitary infrastructure in this context. This method implies overdesign because it considers an equal rainfall affecting the whole micro-basin. Hydrologic simulation is recommended for the final proposal. It is also recommended that the hydrogeology be refined with a borehole exploration plan, because substantial spatial heterogeneity in the subsurface is expected and it may render the recharge flux computation less reliable.

Considering the current flood problems and water shortages that the city faces, this technology appears to be an essential contribution for improving water management: on the one hand, to alleviate the inundations and on the other, to recharge the overexploited aquifers and advance towards a more sustainable supply. We contend that the prevailing situation in the MAG illustrates that better management and engineering can contribute to solving the hydric problems of many other cities in a similar situation. However, it is still necessary to develop a runoff treatment strategy and a management plan to ensure water quality and to avoid negative environmental repercussions. Additionally, constructing an exploratory well, fitted with the relevant instruments of measurement, may provide more precise computations along with the complete stratigraphic profile, permitting calibration of numerical models and more efficient measurements for future interventions.

An interesting correlation exists between reducing over-impacting infrastructure and creating a sustainable and improved quality water supply. A highly beneficial consequence of adapting the suggested solution would mean a much smaller environmental impact due to reduced infrastructure requirement in the frame of a sustainable urban water management strategy.

This approach is suitable in urban areas where land cost tends to be high and aquifers tend to be overexploited. Certain geohydrological conditions are required, such as significant surface runoff, acceptable water quality for infiltration, considerable depth in the phreatic levels, and good subsoil permeability. It may be particularly useful when the aquifer for recharge is alternated with strata of low permeability. It should also be considered that subsurface recharge may be reduced over time due to clogging by fine particles as well as microorganisms, and a maintenance plan is required.

Finally, this recharge plan can support public health, providing a larger quantity of higher quality water available, enhancing the quality of the environment in regularly flooded areas, and contributing to the reduction of respiratory illness, as recharging the aquifer implies fewer temperature fluctuations.

## Figures and Tables

**Figure 1 ijerph-19-03160-f001:**
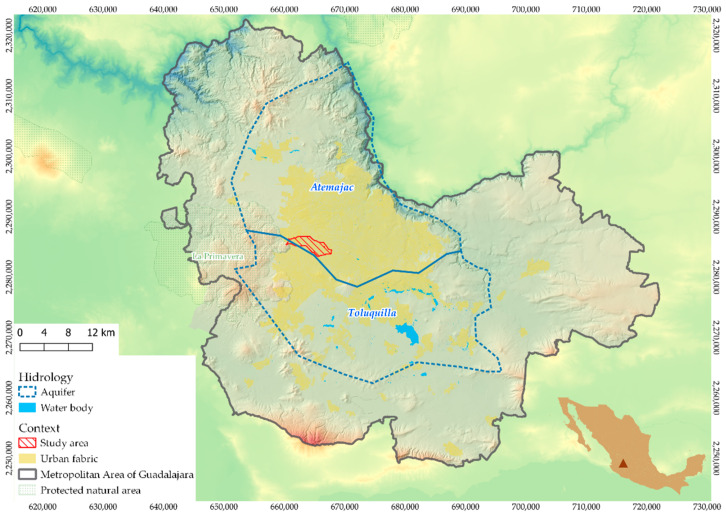
Location of micro-basin El Chicalote in the Metropolitan Area of Guadalajara (MAG).

**Figure 2 ijerph-19-03160-f002:**
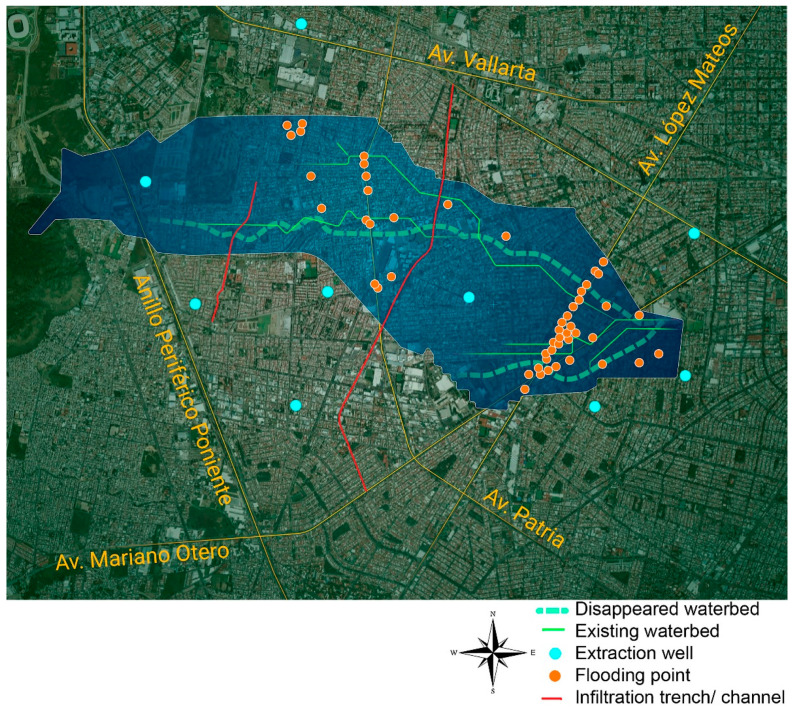
Micro-basin El Chicalote: hydrological context.

**Figure 3 ijerph-19-03160-f003:**
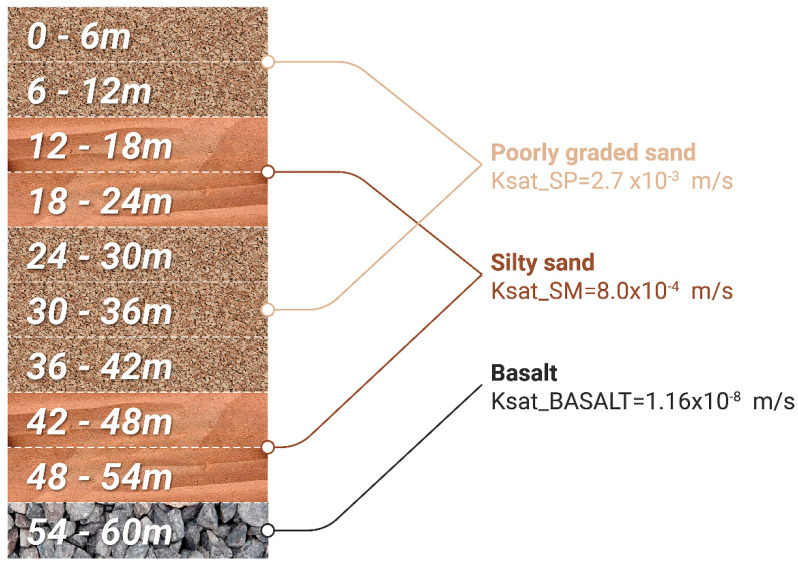
Mean stratigraphic profile in micro-basin of El Chicalote.

**Figure 4 ijerph-19-03160-f004:**
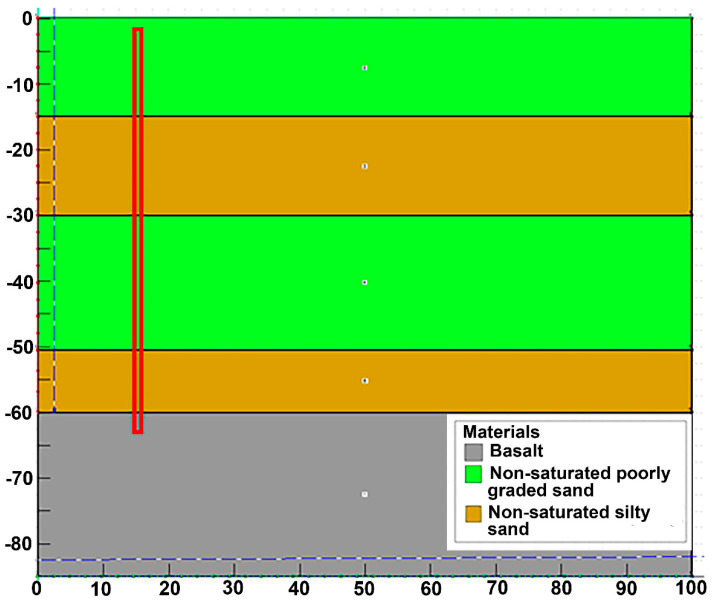
Geometry of the numeric model in SEEP/W.

**Figure 5 ijerph-19-03160-f005:**
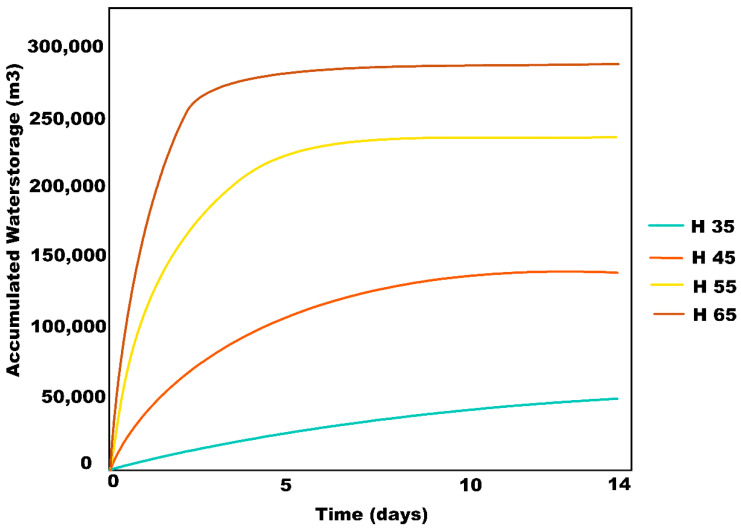
Accumulated water volume for 1 m diameter wells with different depths for 14 days. in SEEP/W.

**Figure 6 ijerph-19-03160-f006:**
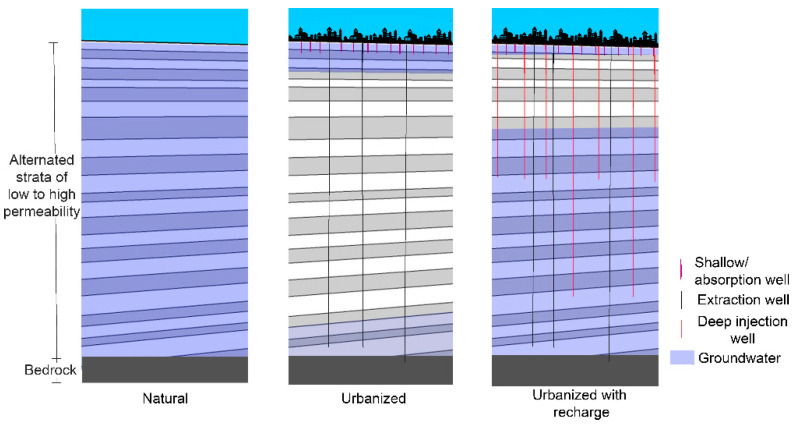
Conceptual scheme.

**Table 1 ijerph-19-03160-t001:** Analyses elements and data gathered for the assessment.

Units of Analyses	Observables	Information Collection Strategy	Source	Analysis Method	Product
Surface hydrology	- Aquifers - Channels - Urban usage water wells - Existing waterbed - Pre-existing waterbed - Terrain slopes - Floods record	Documentary research	[8] [21] [17]	- Map algebra in Geographic Information Systems. - Rational method - Triangular unit hydrograph method	- Selection of a micro-basin with intervention potential - Runoff volumes - Maximum runoff flux
Geohydrology	- Rock stratum depth - Phreatic level depth - Stratigraphy - Permeability coefficients	Documentary research	[19] [22] SPTs performed by geotechnical engineering companies [20]	- Hermeneutical analysis - Triangulation and simple means	- Representative stratigraphic profile of the selected micro-basin. - Strata permeability
Deep Injection well	- Geometric data - Geohydrology	Iterative process	[23]	Finite elements method	Maximum infiltration capacity for wells of different dimensions

**Table 2 ijerph-19-03160-t002:** Maximum infiltration volumetric fluxes for different depths during the first 12 h.

Wells’ Depth (Meters)	Volumetric Flux (m^3^/s)
35	0.42
45	0.92
55	1.06
65	1.85

## Data Availability

The data presented in this study are available on request from the corresponding author. The data are not publicly available due to privacy issues.

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
