# Peer review of "Flood Mitigation in Urban Areas through Deep Aquifer Recharge: The Case of the Metropolitan Area of Guadalajara"

_ijerph, 2022, doi:10.3390/ijerph19063160_

Round 1

Reviewer 1 Report

Dear Authors, 

I had the chance to have a look at your paper. Unfortunately, the level of English is very poor and, as a result, I have to reject your paper, regardless of the scientific content. 

I would recommend to ask someone with high level in English to edit your paper and make it clear and easy for the reader to comprehend.

Kind regards

Author Response

The paper has been reviewed by two English native speakers with scientific experience for a complete check to make it clear and easy for the readers to comprehend.

Reviewer 2 Report

The authors described and performed an analysis of flood mitigation in the city of Guadalajara by recharging deep aquifers. The case is described for real conditions. The methodology is correctly adopted. Three types of analyzes were performed:
1. Hydrological research to determine the amount of run-off that must be consumed.
2. Geohydrological conditions of the subsoil for the assessment of the subsoil's ability to receive runoff.
3. Numerical modelling of infiltration of theoretical wells of various sizes.
The conclusions result from the content of the conducted research. They are clear and easy to read.

Only the discussion contains a few citations from other authors. The authors mention that a similar system works in Dusseldorf and Budapest (line no. 119) - these systems could be briefly described - their differences and similarities to the one presented in this work.

The work is carefully prepared. The drawings are properly prepared.

Author Response

The authors acknowledge the anonymous reviewer for his/her helpful comment.

The Dusseldorf and Budapest systems have been briefly described in the discussion section and compared to the one proposed in the paper. Two new references have been added.

  • Eckert, P; Irmscher, R. Over 130 years of experience with Riverbank Filtration in Düsseldorf, Germany, Journal of Water Supply: Research and Technology, 2006, Vol. 55, Nº 4, 283-291.

  • Grützmacher, G., Hülshoff, I., Wiese, B., Moreau-Le Golvan, Y., Sprenger, C., Lorenzen, G., & Pekdeger, A. Function and relevance of aquifer recharge techniques to enable sustainable water resources management in developing or newly-industrialized countries. In T. van den Hoven and C. Kazner, TECHNEAU: Safe Drinking Water from Source to Tap, 2009, Vol. 8, 121-132. https://doi.org/10.2166/9781780401782

Reviewer 3 Report

General comments

The authors need to specify the link with public health and revise the porosity values reported. Please, follow the specific comments and add the suggested relevant references

Specific comments

Abstract

Lines 14-28. Define how deep, provide values in meters. This detail helps the readers to understand the scale of investigation

Introduction

Lines 37-41. Sentence not backed-up by references. You need to insert these two references that treat problems of aquifer recharge and water scarcity. Note that, the first paper on aquifer recharge treats an analogue scenario with a stratigraphy characterized by clastic deposits and basalts

- Medici, G., Engdahl, N. B., & Langman, J. B. (2021). A basin-scale groundwater flow model of the Columbia Plateau regional aquifer system in the Palouse (USA): Insights for aquifer vulnerability assessment. International Journal of Environmental Research15(2), 299-312.

- Zhang, H., Xu, Y., & Kanyerere, T. (2020). A review of the managed aquifer recharge: Historical development, current situation and perspectives. Physics and Chemistry of the Earth, Parts A/B/C118, 102887.

Line 45. Explain the reason for water scarcity in Mexico. Conditions of semi-aridity and aridity?

Line 70. “Rock layer”. You should use the word “bedrock”

Line 77. “80%” porosity is a value un-real and 40% very high.

Line 78. Please, specify the type of fault. Extensional?

Lines 100-114. Please, specify how deep in terms of meters below the ground

Material and methods

Line 155. Specify soil stratigraphy and provide depth interval

Line 156. Specify, if possible, the geotechnical engineering companies involved in your research

Results

Line 194. Provide detail on how you derived this equation

Line 194. The equation should appear above in the material and methods and then re-called in the results for the derived computations

Line 222 - to the end. You mention “micro-basin”, make sure you define in the introduction definition of the world and scale of investigation

Line 239. Specify method used to extrapolate permeability

Line 303. Work out the value of aquifer recharge in mm/year

Discussion

Lines 310-355. Specify here or in the introduction the link with public health

Lines 310-355. I leave to the authors the possibility to add a conceptual scheme of the take home message of the paper

Line 314. I suggest changing “geology” into “hydrogeological background”

Conclusions

Lines 357-373. Specify if your approach is particularly suitable for one or more hydrological/hydrogeological scenarios

References

Line 392. Add the relevant and recent references suggested above

Figures and tables

Figure 1 and 2. Add the fault that contributes to the recharge?

Author Response

The authors acknowledge the anonymous reviewer for his/her helpful and constructive comments.

The paper has been modified addressing these general comments by attending to the following specific comments and the suggested relevant references have been added.

The detailed answers are listed in the attached file:

Round 2

Reviewer 1 Report

Dear Authors,

I find that you improved the initial version of the paper with regards to the language. Some additional comments about it follow.

I would recommend to use the term "water" instead of "hydric" (e.g. "water resources" instead of "hydric resources")

The Introduction needs severe rearrangement since many of the things that are mentioned there (e.g. L74-89) belong to a "Study area" section that is missing and can be placed before the "Materials and Methods"

Table 1: Use the margins in the Table, it will make it easier for the reader to understand

Figures 1 and 2: Using white backgrounds in the legends will make it easier to read

Section 3.3: Since you are using numerical simulations in the study it is necessary to provide information about the characteristics of the model you built, the boundary conditions used, the spatial and temporal discretization etc.

You consistently refer to permeability and give values in m/s which is the unit for hydraulic conductivity. As yourself mention in the equation in L387, the hydraulic conductivity is calculated using the permeability. Please clarify

Author Response

Answers to reviewer’s comments and suggestions

REVIEWER 1

The authors acknowledge the anonymous reviewer for his/her helpful comments.

Comment 1

I would recommend to use the term "water" instead of "hydric" (e.g. "water resources" instead of "hydric resources")

Answer 1

The term "water" has been used instead of "hydric" when linked to resources as recommended.

Section

Introduction

Comment 2

The Introduction needs severe rearrangement since many of the things that are mentioned there (e.g. L74-89) belong to a "Study area" section that is missing and can be placed before the "Materials and Methods"

Answer 2

A new “Study Area” section has been placed before the "Materials and Methods" and the Introduction has been rearranged, moving the appropriate contents to the new section. The subsequent sections have been renumbered and the references reordered.

Section

Introduction

Comment 3

Table 1: Use the margins in the Table, it will make it easier for the reader to understand

Answer 3

Margins have been used in the Table 1 as suggested.

Section

Materials and Methods

Comment 4

Figures 1 and 2: Using white backgrounds in the legends will make it easier to read

Answer 4

Figures 1 and 2 have been edited using white backgrounds in the legends as recommended.

Sections

Study Area and Results

Comment 5

Section 3.3: Since you are using numerical simulations in the study it is necessary to provide information about the characteristics of the model you built, the boundary conditions used, the spatial and temporal discretization etc.

Answer 5

Information about the characteristics of the built model, the boundary conditions used, the spatial and temporal discretization, etc. have been provided in new lines 312-332.

Section

Results

Comment 6

You consistently refer to permeability and give values in m/s which is the unit for hydraulic conductivity. As yourself mention in the equation in L387, the hydraulic conductivity is calculated using the permeability. Please clarify

Answer 6

These terms have been corrected in new lines 295-300, and 354 to clarify them.

In addition, aquifer recharge figures have been corrected in new lines 379-384.

Section

Results